# Rho-Proteins and Downstream Pathways as Potential Targets in Sepsis and Septic Shock: What Have We Learned from Basic Research

**DOI:** 10.3390/cells10081844

**Published:** 2021-07-21

**Authors:** Maria Luísa da Silveira Hahmeyer, José Eduardo da Silva-Santos

**Affiliations:** Laboratory of Cardiovascular Biology, Department of Pharmacology, Universidade Federal de Santa Catarina, Florianópolis 88040-900, SC, Brazil; mlhahmeyer@gmail.com

**Keywords:** endotoxemia, endotoxic shock, organ dysfunction, Ras proteins, small GTPases

## Abstract

Sepsis and septic shock are associated with acute and sustained impairment in the function of the cardiovascular system, kidneys, lungs, liver, and brain, among others. Despite the significant advances in prevention and treatment, sepsis and septic shock sepsis remain global health problems with elevated mortality rates. Rho proteins can interact with a considerable number of targets, directly affecting cellular contractility, actin filament assembly and growing, cell motility and migration, cytoskeleton rearrangement, and actin polymerization, physiological functions that are intensively impaired during inflammatory conditions, such as the one that occurs in sepsis. In the last few decades, Rho proteins and their downstream pathways have been investigated in sepsis-associated experimental models. The most frequently used experimental design included the exposure to bacterial lipopolysaccharide (LPS), in both in vitro and in vivo approaches, but experiments using the cecal ligation and puncture (CLP) model of sepsis have also been performed. The findings described in this review indicate that Rho proteins, mainly RhoA and Rac1, are associated with the development of crucial sepsis-associated dysfunction in different systems and cells, including the endothelium, vessels, and heart. Notably, the data found in the literature suggest that either the inhibition or activation of Rho proteins and associated pathways might be desirable in sepsis and septic shock, accordingly with the cellular system evaluated. This review included the main findings, relevance, and limitations of the current knowledge connecting Rho proteins and sepsis-associated experimental models.

## 1. Introduction

Sepsis is defined as life-threatening organ dysfunction caused by a dysregulated host response to infection, as introduced by the Third International Consensus Definitions for Sepsis and Septic Shock in 2016 [1]. The clinical management and criteria for the early diagnosis of sepsis and septic shock have been systematically reviewed since the 1990s, implementing significant advances in prevention, treatment, and survival rates associated with these conditions. Nevertheless, sepsis remains a global health problem and a significant cause of mortality, accounting for around 20% of worldwide deaths in 2017 [2]. The knowledge regarding the onset and prognosis of sepsis remains under ongoing discussion, but it depends on associated risk factors and underlying conditions. The elderly, young children, and immunocompromised individuals are examples of people included in risk groups, and serum levels of pro-inflammatory and anti-inflammatory cytokines have been explored to predict fatal outcomes in septic patients [3]. Indeed, one of the primary reasons sepsis is hard to be predicted and treated is its complex pathophysiology, which includes almost all known inflammatory mediators, differentially found in distinct tissues and stages of the disease. The acute or sustained impairment in vital systems can compromise the kidneys, lungs, liver, coagulation, and central nervous system, among others. Notably, the cardiovascular system is affected from early to late stages of sepsis, and the dysfunction in several organs may be at least partially resulted from reduced blood perfusion and augmented vascular permeability, which are hallmarks with putative clinical value for the prognosis of this condition [4]. In the vascular system, high levels of reactive oxygen species (ROS), including nitric oxide (NO), are produced, beginning in the very early stages of sepsis, triggering endothelial dysfunction. The unbalanced production of endothelial mediators and exacerbated levels of inflammatory cytokines contribute to persistent hypotension, leading to the inadequate blood supply and poor tissue perfusion, reducing O_2_ distribution. Sepsis progresses to septic shock when physiological mechanisms and fluid resuscitation are unable to restore blood pressure, the patient presents hyperlactatemia (>2 mmol/L), and vasopressor drugs are required to maintain the minimum 65 mm Hg mean arterial pressure. In more severe stages, the vascular system became hyporeactive to vasoconstrictors, and the desired arterial pressure cannot be targeted even when high doses of pressor agents are used [1]. For this, components of the cardiovascular system may be considered crucial targets to prevent sepsis-associated organ dysfunction.

The discovery of small G proteins, followed by the description of their widespread distribution and biological properties, raised an unexplored field for drug targets with putative importance in several diseases, including sepsis and septic shock. The Rho subfamily is part of the Ras superfamily of small GTPases, and is composed of 20 members in mammalians, further subdivided into eight different subclasses (i.e., Rho, Rac, Cdc42). Rho proteins have been broadly studied in several areas, such as the regulation of cell migration [5], calcium sensitization, and the maintenance of smooth muscle tone, i.e., [6,7], differentiation [8], cell growth and apoptosis [9], regeneration [10], focal adhesion [11,12] and polarity [13,14], cytokinesis [15,16], and membrane trafficking and ruffling [17,18], among others. Recent studies have also suggested that Rho-GTPase signaling pathways crosstalk with each other and are influenced by cellular mechanics, leading to the self-organization of several dynamic cellular processes (for reviews, see [19,20,21]). The involvement of Rho proteins in so many cellular functions explains why these small G proteins and their downstream pathways have a pivotal role in both physiological and pathological conditions.

Rho proteins are small GTPases that switch between active and inactive stats, dependent on GTP or GDP biding, respectively, a process regulated by guanine nucleotide exchange factors (GEFs), GTPase-activating proteins (GAPs), and guanine nucleotide dissociation inhibitors (GDIs). Similar to classical heterotrimeric G proteins, Rho proteins are also subjected to activation by different subtypes of transmembrane receptors classified as G protein-coupled receptors (GPCRs). However, unlike heterotrimeric G proteins, the mechanisms throughout GPCRs activate Rho proteins are indirect and dependent on GEFs. Moreover, it mainly includes receptors coupled to G proteins containing the α_q_ and α_12/13_ subunits, i.e., [22,23]. Several receptors with key regulatory effects in the vascular smooth muscle cells (i.e., α_1_-adrenoceptors and angiotensin II AT1 receptors), endothelium (i.e., bradykinin B2 receptors), and platelets (i.e., protease-activated receptor-1, PAR-1) are coupled to Gα_q_ and Gα_12/13_ proteins. Importantly, these GPCRs have been used or at least investigated as drug targets in the management of sepsis. Besides, Rho proteins are also reached by other classes of receptors, such as integrin [24] and peroxisome proliferator-activated receptors [25], reinforcing their multiple regulatory roles in cell signaling.

Rho proteins can interact with a considerable number of targets, regulating multiple processes that are all subjected to pathological or adaptive changes with functional relevance during inflammatory processes, such as the one that occurs in septic insults. We aim to provide the readers with a comprehensive overview to allow the understanding of how the study of Rho proteins and their downstream pathways, particularly the Rho-associated coiled-coil-containing protein kinase (Rho-kinase or ROCK, further classified as subtypes I and II), are directly or indirectly involved in different signs or symptoms of this disease, and perhaps why it can be a source of innovative targets for the clinical management of sepsis and septic shock. As summarized in Figure 1 and explored ahead, Rho proteins and Rho-associated targets are involved in the main signs of sepsis and septic shock.

## 2. Experimental Models Used to Study Sepsis-Associated Dysfunctions

Sepsis and septic shock are systemic conditions that gradually affect all vital functions in humans and animals. The complexity of such effects, the intensive supporting care, fluid replacement, and multiple drugs used, including but not limited to antibiotic therapy, vasopressor, and inotropic agents, creates a scenario that is virtually impossible to be entirely replicated in laboratory studies. Notably, our knowledge about how sepsis impairs human physiology has been built under the light of approaches that attempt to reproduce at least part of the infection or inflammatory process that occurs in this condition. For instance, although it cannot be defined as experimental models of sepsis, proinflammatory cytokines, mainly the tumor necrosis factor (TNF)-α and interleukin (IL)-1β have been used to create the inflammatory environment in studies focused on sepsis.

Bacterial lipopolysaccharide (LPS) has been used in in vitro and in vivo studies. It can reproduce much of the inflammation-associated responses found in cells, tissues, and entire animals, such as the production of proinflammatory cytokines, impaired metabolism or muscle contractility, hypotension, and low blood perfusion. The main limitations of LPS as an experimental model for the study of sepsis-associated events are the lack of an ongoing infection, the self-limited duration of the effect when used in in vivo studies, and its action centered on the activation of the Toll-like receptor (TLR) 4. Although the status of LPS as a reliable experimental model of sepsis remains under discussion [26], the challenge with LPS allows the investigation of sepsis-associated responses in cultured cells under very controlled conditions. Indeed, it is widely used and can be considered an excellent choice for the study of cell signaling intracellular processes putatively impaired by sepsis.

Additional models for the study of sepsis include the cecal ligation and puncture (CLP), the intraabdominal injection of fecal pellets, and the administration of pathogenic microorganisms [27,28,29,30]. Among these, the CLP model, which consists of a controlled injury in the cecum that generates polymicrobial sepsis, has been described as the golden model in terms of similarity with the temporal course and organ dysfunction that occurs in septic shock in humans [26,28].

In this review, we discuss the results obtained from in vitro and in vivo studies that explored the behavior of Rho proteins and related intracellular signaling pathways in experimental conditions potentially associated with sepsis, mainly LPS challenge and stimulation with proinflammatory cytokines. Whenever performed, studies using the induced sepsis model were included in our description.

## 3. Rho Proteins and Their Impact on Endothelial Function in Sepsis-Related Experimental Approaches

The relationship between changes in endothelial function and sepsis is narrow. The exacerbated inflammation that occurs in sepsis leads to endothelial dysfunction, which in this condition is characterized by the unbalanced production of endothelial factors crucial for several physiological events, including the maintenance of the own endothelial biology, preservation of the endothelial barrier function, prevention of blood clotting, and the regulation of the vascular tone. As summarized in Table 1, the involvement of Rho proteins in sepsis-associated endothelial dysfunction has been continuously investigated.

One of the first pieces of evidence that the RhoA/ROCK pathway could play a role in the pathophysiology of sepsis was the demonstration of reduced activity of myosin light chain phosphatases (MLCP) and enhanced myosin light chain (MLC) phosphorylation in human endothelial cells incubated with LPS, which was counteracted by inhibitors of ROCK and cAMP [31]. This augmented activation of Rho components can contribute to endothelial contraction and vascular leakage in sepsis, as shown in both LPS-treated and CLP-subjected animals [32,33,34,35,36]. Among the mechanisms associating Rho protein and endothelial barrier disruption, the activation of RhoA and/or ROCK in response to LPS were involved in in vivo leukocyte adhesion in femoral [32] and hepatic [37] arteries of mice, the loss of VE-cadherin function in endothelial cells [38,39,40], increased expression of adhesion molecules [40], and neutrophil–endothelial adhesion [41].

RhoA has previously been described as susceptible to inhibitory regulation by Rac [42]. Thus, the delicate balance between the antagonistic effects of Rac and RhoA activities is a potential target for regulating the endothelium barrier function in sepsis, as suggested by the anti-permeability effect of angiopoietin-1 in both endothelial cells and mice exposed to LPS [43]. Nevertheless, this is a multicomplex process that involves the entire contractile machinery in the endothelial cells, as explored by Bogatcheva and co-workers, who also found that the inhibition of ROCK can either improve or worsen the barrier function of human lung microvascular endothelial cells [44]. Accordingly, the activation of RhoA by the active biolipid sphingosine-1-phosphate was suggested as an important way to counteract pericyte loss and improve barrier function in endothelial cells subjected to LPS [45]. Moreover, both the activation and inactivation of endothelial Rho by *Clostridium limosum* exoenzyme and *Escherichia coli* cytotoxic necrotizing factor CNF1, respectively, improved the endothelial barrier, reducing leukocyte migration [46]. Additionally, Adamson and co-workers described that the inhibition of ROCK reduced the basal unstimulated vascular permeability in rat venular microvessels of the mesentery but failed to prevent the transient permeability induced by bradykinin and platelet-activating factor [47]. Besides, augmented levels of active Rac1 and Cdc43, and unaltered RhoA, were detected in CNF1-stimulated myocardial endothelial cells [48].

Despite the few studies detailing how LPS or sepsis modulate Rho proteins in endothelial cells, the effects of endogenous molecules that are increased under LPS or septic insults, such as thrombin [49], heparin-binding protein [50], and the heat-shock protein 90 [51], have been linked with Rho-dependent mechanisms. Additionally, there is substantial evidence that the engagement of Rho proteins in endothelial barrier regulation can be dependent on the kind of vessel or cell evaluated [52,53,54]. For instance, the activation of Rho was necessary for TNF-α-mediated barrier dysfunction in human lung microvascular endothelial cells [55] and umbilical vein endothelial cells [56], but only Rac accounts for this effect in human dermal microvascular endothelial cells [57]. Indeed, the role of Rho proteins on endothelial hyperpermeability depends on both the inflammatory mediator [58] or the time point evaluated [59]. In any case, endothelial dysfunction has a putative causal relationship with infections and inflammatory processes, which also can be modulated by Rho proteins [60,61]. In this way, the activation of the TLR 4 by LPS upregulated the guanine–nucleotide exchange factor GEF-H1 and increased the activity of RhoA in human umbilical vein endothelial cells, and this process was characterized as an upstream step for nuclear factor kappa B (NF-κB) transactivation and interleukin (IL)-8 expression [62]. Interestingly, the activation of GEF-H1 and NF-κB were also found in *Staphylococcus aureus*-induced endothelial barrier dysfunction [63]. Furthermore, the inhibition of ROCK significantly reduced LPS-increased inflammatory cytokines IL-1β and IL-6 in human lung microvascular endothelial cells, at least partly via NF-κB inhibition [64]. Similar findings associating the RhoA pathway and the NF-κB activity with LPS-induced endothelial hyperpermeability were also previously described in mouse brain-derived microvascular endothelial cells [65].

Mediators produced by the endothelial cells, including NO, metabolites of arachidonic acid, and ROS, are often involved in the control of vascular tone. However, they are also crucial to regulate the endothelial barrier, platelet aggregation, and the expression of adhesion molecules, among others. Transgenic mice expressing CYP2J2, the cytochrome P450 epoxygenase 2J2 that produces epoxyeicosatrienoic acids, presented reduced mortality rates when treated with lethal doses of LPS. This protection was explained by the ability of epoxyeicosatrienoic acids to inhibit the generation of ROS and the subsequent activation of the RhoA/ROCK pathway in endothelial cells, preventing MLC phosphorylation and improving the endothelial barrier [66].

The widespread presence of the nitric oxide synthase (NOS) among mammalian cells makes NO a key mediator in several systems. Notwithstanding the multiple physiological roles of NO, the nature of its effects on cellular biology are entirely dependent on the amounts produced. Along with inflammatory burst that occurs in sepsis, the overexpression of the inducible isoform of NOS (iNOS) exacerbates the vasodilatory and hypotensive effects generated by NO, contributing to the reduced blood flow and the organ damage seen in this condition. Although strategies capable of preventing or reversing the harmful effects of NO remain desirable, maintaining the activity of endothelial NOS (eNOS) might be essential for vascular recovery in sepsis. Notably, the administration of the ROCK inhibitor fasudil prevented the overexpression of iNOS and increased the levels of eNOS in mesenteric endothelial cells from rats treated with LPS, an effect accompanied by a reduction in both macromolecular leak and leukocyte adhesion in mesenteric arteries [67]. The dependence of eNOS for the anti-inflammatory and antiapoptotic effects obtained after the inhibition of the RhoA/ROCK pathway in LPS-treated endothelial cells was also described [68]. Endothelium-derived mediators keep an orchestrated equilibrium to maintain vascular functions. Under physiological conditions, endothelin-1, a potent vasoconstrictor, also promotes the vascular release of NO [69,70]. This function can be disrupted under inflammatory stimuli, such as LPS, but the pharmacological inhibition of the RhoA/ROCK pathway was able to bring back endothelin-1-mediated eNOS activity [71]. Regarding advances into the molecular regulatory mechanisms on Rho proteins, peroxynitrite-mediated RhoA and Rac1 nitration [72,73] were proposed as a key event driving the increment and inhibition of RhoA and Rac1 activities, respectively, which contributes to LPS-induced endothelial barrier disruption.

Regardless of the missing points that would enhance our comprehension about how the function of Rho proteins and its up or downstream targets are modulated in endothelial cells during septic insults, these small G proteins have been consistently included in the physiological regulation of endothelial structure and function in different vascular systems [74,75,76,77,78,79,80,81,82,83,84,85,86,87]. In summary, the balanced activity of Rho proteins has been reported as mandatory to maintain several aspects of endothelial biology, including F-actin stabilization, contractility, and functional tight junctions (Figure 2A). This balance is disrupted when endothelial cells are subjected to a proinflammatory environment such as the one found in sepsis, resulting, for instance, in the reduced activity of Rac1 and augmented activity of the RhoA pathway, which contribute to endothelial dysfunction (Figure 2B). Notably, the direct or indirect modulation of these proteins by GEFs, i.e., [88,89], chemical compounds, i.e., [90,91,92], endogenous mediators, i.e., [45,56,93,94], or even RNA manipulation, i.e., [62,63,95], may provide additional insights about the putative role of Rho proteins in the endothelial function in sepsis (see Table 1).

## 4. Rho Proteins and Their Impact on the Vascular Function in Sepsis-Related Experimental Approaches

The RhoA pathway is particularly important in vascular tone regulation. Briefly, the stimulation of several GPCRs in smooth muscle cells leads to phospholipase C activation, the production of phosphatidylinositol 1,4,5-trisphosphate, and Ca^2+^ release from the sarcoplasmic reticulum. The augmentation of cytosolic Ca^2+^ free levels enables calmodulin to interact with myosin light chain kinase (MLCK), triggering its activity. The more MLCK is active, the more it phosphorylates the myosin light chain (MLC), which increases its interaction with actin, leading to greater contractility. This process is physiologically contained by myosin light chain phosphatase (MLCP), which dephosphorylates the MLC, reducing the contractile tone. The receptor-mediated activation of RhoA proteins increases the activity of ROCK, which in turn inhibits MLCP. It maintains the cell more susceptible to calcium-mediated MLCK activation. For this, the RhoA/ROCK pathway is defined as a pro-contraction system and a calcium sensitization route in cell signaling. This physiological role of Rho proteins in vascular muscle cells and its relevance for the maintenance of systemic arterial pressure is illustrated in Figure 3A. It is important to note that tone regulation is not the only physiological function directly modulated by Rho proteins in vascular smooth muscle cells. However, perhaps because the augmented activation of this pathway plays a role in developing hypertensive disorders [7], the most investigated aspect of Rho components in the septic vascular smooth muscle has been its involvement in vascular contractile dysfunction, as can be seen in Table 2.

The incubation of LPS in vascular preparations maintained in organ baths for assessing contractile responses usually results in a pattern of vascular hyporeactivity to vasoactive drugs that resemble the vasoplegia found in septic patients. Using rat pulmonary arteries, Boer and co-workers demonstrated that LPS-induced vascular dysfunction is also characterized by augmented compliance, as evaluated by the diameter of arteries subjected to cumulative stretch, a condition accompanied by a disordered distribution of F-actin in the smooth vascular cells. This disassembly of the F-actin fiber was reproduced by the pharmacological inhibition of ROCK in control vessels (not exposed to LPS) and, most importantly, was preventable by the activation of RhoA in LPS-exposed pulmonary arteries [96].

LPS was found as a negative modulator of the RhoA/ROCK pathway in several studies, i.e., [97]. However, augmented levels of ROCK were also found in endotoxemia, simultaneously to the potentiated contractile responses to endothelin-1, as described in the superior mesenteric artery of rats at 6 h after LPS administration [98]. Despite the vascular system being treated as a unit, each vascular bed behaves differently depending on the organ evaluated. These differences seem to be particularly important for experimental models of sepsis since the pattern of responses evoked by vasoactive agents depends on the vasculature and time of evaluation [99]. Accordingly, endothelin-1-induced contractile responses and the phosphorylation of MLC were reduced in rat aortic rings at 20 h after incubation with LPS, but these changes were independent of changes in the RhoA/ROCK pathway [100]. Additionally, the expression levels of Rho pathway components were increased in resistance mesenteric arteries from LPS-treated rats at both early (6 h) and late (24 h) periods of endotoxemia, although the vessels were hyporeactive to phenylephrine and more sensitive to ROCK inhibition [101]. Similar findings were described in rat aortic rings at 6 h after LPS administration [102].

Although precise mechanisms remain undefined, inflammatory cytokines, high levels of NO, produced mainly by the iNOS isoform, the impaired balance of ROS, and the modulation of downstream targets, such as the soluble guanylate cyclase, potassium and calcium channels, have been described as the main factors responsible for the cardiovascular dysfunction in sepsis. The exposure to either IL-1β and TNF-α caused a concentration-dependent reduction in calcium-induced contraction and decreased the levels of the phosphorylated MLCP in the superior mesenteric artery of rabbits, suggesting that both cytokines can lead to reduced ROCK activity [103,104]. We have previously demonstrated that the blocking of NO production and the inhibition of the soluble guanylate cyclase can restore the vascular reactivity and increase the phosphorylation of the MLCP in resistance mesenteric arteries from endotoxemic rats [101]. In fact, at least part of the relaxation induced by NO in both vascular and non-vascular smooth muscles might be due to its inhibitory action on the RhoA/ROCK pathway [105,106]. On the other hand, Rho proteins can also regulate the expression of iNOS in vascular smooth muscle cells stimulated by IL-1β [97].

Rho-associated signaling pathways were also evaluated in animals subjected to the CLP model, and the studies reinforce the idea that the role of Rho proteins and its downstream targets in the pathogenesis of the vascular dysfunction in sepsis also depends on the time, the vascular system, and the vasoactive agents evaluated. For instance, at five days after the septic insult induced by CLP, mice femoral arteries presented unaltered contractile responses to phenylephrine and norepinephrine, but these arteries were less reactive to the thromboxane receptor agonist U46619 and displayed reduced levels of ROCK-dependent phosphorylation of MLCP [107]. On the other hand, aortas obtained from rats at 60 days after the CLP surgery showed augmented expression levels of RhoA and ROCK and presented enhanced contractile responses to angiotensin II [108], suggesting that Rho proteins and calcium sensitization can be involved in the development of late cardiovascular diseases among patients who survive sepsis. The vasoconstrictor effects of arginine vasopressin and terlipressin, two selective agonists of V1a receptors, a GPCR that stimulates both phospholipase C and RhoA proteins, were also potentiated in the superior mesenteric arteries of both LPS-treated rabbits and CLP-subjected rats, in a way entirely dependent on ROCK-mediated MLCP phosphorylation [104]. Interestingly, we did previously demonstrate that the enhanced activation of Rho-kinase by vasopressin in the renal vascular bed contributes to the maintenance of the pressor effects mediated by V1a receptors during endotoxemic shock in rats [109].

Understanding the pathophysiological aspects involved in the vascular component during septic shock may allow the development of efficient clinical strategies for the management of severe hypotension and poor blood perfusion, which end up triggering multiple organ failure and high rates of lethality in this syndrome. Unlike endothelial cells, from the prism of vascular contractility, most of the experimental studies indicate that sepsis is associated with a depressed functionality of the RhoA/ROCK pathway (Figure 3B), making Rho proteins valuable target candidates (see Table 2 for an overview of the main findings in this topic). For instance, the indirect pharmacological modulation of ROCK by the flavonoid oroxylin-A proved to be able to restore the LPS-induced suppression of RhoA activity in arteries from rats [110]. Nonetheless, the administration of the selective ROCK inhibitor fasudil to rats subjected to the CLP surgery ameliorated acute lung injury, improving several systemic biochemical and inflammatory markers of sepsis severity, including the systemic arterial pressure [35].

## 5. Rho Proteins and Their Impact on the Heart Function in Sepsis-Related Experimental Approaches

The pump heart function is crucial for cardiovascular homeostasis and is also impaired in sepsis, as characterized by reduced left ventricular contractility and deficient relaxation in humans [118,119] and animals, i.e., [120,121]. Decreased responsiveness to sympathetic regulation, increased levels of proinflammatory substances, oxidative stress, impaired endothelial function, augmented migration of leukocytes, and mitochondrial dysfunction, among others, have been associated with impaired myocardial contractility, impaired coronary perfusion, and heart arrhythmia in sepsis and septic shock (for review see [122]). Notwithstanding, the detailed interplay between these multiple events remains to be elucidated. Small G proteins are also present in the healthy and diseased hearts, and their multiple functions include the regulation of apoptosis, gene expression, intercellular communication, hypertrophy, and cardiac remodeling, i.e., [123,124,125,126,127,128,129,130,131].

In recent years, Rho proteins, mainly RhoA and Rac1, have also been involved in sepsis-induced cardiac dysfunction. Soliman and co-workers showed that the levels of RhoA increased in rat ventricular cardiomyocytes after LPS-induced iNOS expression, a process fully reproducible when the cells were incubated with a NO donor, without LPS, indicating that high amounts of NO, such as the amount that occurs in sepsis, can upregulate the RhoA protein in isolated cardiomyocytes [111]. The systemic administration or in vitro incubation of LPS also modulated the activity of Rac1 in the heart and cardiomyocytes of mice [112,114]. Moreover, the use of knockout mice indicated that Rac1 is directly involved in LPS-induced TNF-α expression, the activation of NADPH oxidase, ROS production, and the stimulation of ERK1/2 and p38 MAPK pathways [112,114]. Importantly, the absence of Rac1 also reduced the loss of cardiac contractility found in endotoxemic animals [114]. Interestingly, at 4 h after the administration of LPS, Rac1 activity was higher in the heart of males than in cardiac samples from female mice, and this difference was abolished when the male mice were pretreated with 17β-estradiol [112]. If, on the one hand, the activity of Rac1 appears to explain at least part of the cardiac dysfunction in sepsis, on the other hand, the activation of Ras proteins and downstream ERK signaling pathways were described as the putative mechanism of cardioprotection promoted by TLR 9 in mice subjected to the CLP surgery [132].

Even though RhoA and ROCK appear to have minor effects on calcium sensitization processes in the cardiac muscle, this pathway has also been directly and indirectly involved in the pathophysiology of sepsis-induced cardiac dysfunction. For instance, the administration of fasudil reduced biochemical markers of inflammation and oxidative stress, improved the mitochondrial dynamics limiting mitochondrial fission and the phosphorylation of dynamin-related protein-1 (Drp1), and ameliorated the left ventricular function of hearts from endotoxemic mice [113]. Interestingly, the administration of LPS in rats and the incubation of cardiomyocytes with either LPS or TNF-α increased mitochondrial Drp1, an effect associated with augmented levels of RhoA in cells and fully inhibited by ROCK inhibition [115]. In fact, looking from the prism of cardiac dysfunction, the inactivation of the RhoA/ROCK pathway appears to be desirable and has been associated with beneficial effects (the last part of Table 2 also provides the main findings associating Rho proteins and the cardiac function in sepsis-associated models). For instance, neuregulin-1, a member of the epidermal growth factor family involved in cardiovascular function and disorders [133,134,135], attenuated the percentage of apoptotic cardiomyocytes and reduced the depression of heart function induced by LPS administration in rats, matching its ability to prevent the LPS-increased expression of RhoA and ROCK in cardiac cells [116]. However, when administered in CLP-subjected rats, intermedin 1–53, a recently discovered member of the calcitonin gene-related peptide superfamily with biological effects on heart function (for review, see [136]) prevented hypotension, improved blood flow perfusion in vital organs, and avoided the loss of contractile responses by the cardiac papillary muscle, with the last effect being associated with high levels of phosphorylation of troponin inhibitory subunits. Importantly, the authors suggested that the protective effects of intermedin 1–53 on cardiac function were significantly reduced in animals that also received the ROCK inhibitor Y-27632 [117].

## 6. Rho Proteins and Their Impact on Sepsis Outside the Cardiovascular System

Multiple organ failure is the principal outcome of septic shock [137]. Often, even those organs that are not the source of the disease are affected by the systemic inflammatory response. The kidneys, lungs, liver, gastrointestinal system, and brain are the most frequently affected throughout sepsis. It is well accepted that much of the existing damage occurs, at least in part, as a consequence of local vascular changes in response to inflammatory mediators, poor blood perfusion, and the disruption of the endothelial barrier. For instance, vascular inflammation can lead to a harmful low blood flow in the brain with acute consequences, as characterized by delirium in humans [138]. However, cognitive impairment has been described as a consequence of cerebral injury in sepsis-surviving animals [139,140,141]. The activation of RhoA was demonstrated as a key event for endothelial dysfunction in brain microvascular cell lines after incubation with LPS [65,88]. Moreover, the repeated administration of ROCK inhibitors for seven days resulted in a dose-dependent reduction in the amounts of various inflammatory mediators in the brains of rats subjected to CLP and improved the performance of the animals in cognitive tasks [139]. Interestingly, a similar improvement in learning and memory was found in CLP-subjected mice treated with the sesquiterpene β-elemene, which was associated with reduced levels of brain inflammation and decreased Rac1 activity in the mouse hippocampus [142].

Considering that most sepsis cases begin due to pneumonia [143], the lungs suffer the impact of exacerbated inflammatory processes starting in the initial stages of sepsis. Sepsis draws lungs’ hyperpermeability and consequent edema, which culminate in respiratory failure. The RhoA/ROCK pathway activation seems to be important in lung inflammation, since the pharmacological inhibition of ROCK decreases neutrophil migration and lung edema in experimental models of sepsis [35,144,145,146,147]. Besides, researchers found increased levels of apoptosis in septic pulmonary tissue and pulmonary endothelial cells, and ROCK inhibition can prevent this situation [148,149]. Chen and co-workers suggested that the inhibition of the RhoA/ROCK pathway is an essential part of the mechanisms by which the coumarin compound esculetin improves LPS-induced inflammatory damage in the lung epithelium [150]. In recent years, microRNAs have been introduced as an experimental strategy to modulate RhoA/ROCK-mediated events. The administration of microRNA can selectively regulate the expression of diverse mRNA, modulating cell responses. In this way, the use of microRNA in sepsis and endotoxemia can reduce pulmonary inflammation by inhibiting RhoA/ROCK activation [151,152].

Another hallmark of sepsis is the disseminated intravascular coagulation and coagulopathy [153]. Interestingly, anticoagulant effects can account for at least part of the beneficial effects of ROCK inhibitors against LPS-induced lung injury [64]. It was demonstrated that unfractionated heparin reduces RhoA-GTP levels in the lungs, improving LPS-induced pulmonary injury [34]. This finding suggests that the beneficial effects of heparin in sepsis can go further than its anticoagulant properties. Platelets are critical players in hemostasis and are among the first blood cells to accumulate at an injured site. Thus, platelets can be excessively activated under endothelial malfunction, such as the one that occurs in sepsis. The overconsumption of platelets in sepsis can aggravate endothelial dysfunction and lead to thrombocytopenia, a condition associated with bleeding, organ dysfunction, and poor prognosis (for review, see [154]). Notably, the small Rho proteins can, directly and indirectly, be involved in the regulation of platelet aggregation [17,155,156]. The relevance of RhoA-mediated signaling transduction for the activity of platelets in sepsis has been suggested in studies involving both animals and human subjects. In animals, it was demonstrated that fasudil administration to mice treated with LPS reduced the rolling and adhesion of platelets to the endothelium of femoral arteries [157]. In humans, it was found that the incubation with Y-27632 was able to significantly reduce arachidonic acid-induced platelet aggregation in blood samples obtained from both septic and non-septic patients, revealing that the RhoA/ROCK pathways remain functional in platelets during the septic condition [158].

Although scarcely investigated, septic shock and experimental models of sepsis can result in a dysfunctional gastrointestinal system [159,160]. The regulation of the epithelial barrier and calcium sensitization by Rho proteins have been involved in both physiological and inflammation-associated disorders in the intestinal system [161,162,163,164,165,166,167]. The enhanced activity of RhoA was described as a determinant step for diminished epithelial resistance, leading to reduced levels of occludin and E-cadherin found in Caco-2 cells previously incubated with LPS [68]. Interestingly, the severe damage of intestinal epithelial tight junctions induced by the association of D-galactosamine and LPS in mice was prevented by ROCK inhibition, which also restored the occludin levels in the animals [168]. In guinea pigs, the intravenous injection of LPS resulted in spontaneous relaxation of the colonic muscle, a response that almost vanished with fasudil [36]. The treatment with Y-27632 also reduced apoptotic cell levels in the ileum of infant rats after the LPS challenge [169]. In addition, the degree of intestinal injury, levels of tight junction proteins, and markers of the inflammatory response were all improved in CLP-subjected mice treated with the flavanone glycoside naringin, an effect associated with the inhibition of RhoA and ROCK activities in the ileum [170]. Moreover, pleiotropic actions of simvastatin include the downregulation of both RhoA and ROCK in the intestinal tissue of CLP-subjected rats, and this effect may explain, at least in part, how simvastatin prevents the intestinal barrier disruption in septic animals [171].

The maintenance of liver function is critical for survival in sepsis (for review, see [172]). Thus, experimental studies involving the systemic effects of sepsis frequently include the measurement of hepatic biomarkers. Nevertheless, few studies have explored the biological role and the impact of the pharmacological modulation of Rho proteins in liver function in experimental models of sepsis. As described for different tissues, high levels of RhoA and Ras proteins were found in the liver of mice subjected to CLP [173]. Both fasudil and Y-27632 can prevent LPS-induced liver dysfunction in mice, resulting in reduced apoptosis, diminished rolling, the adhesion and accumulation of neutrophils, and improved blood perfusion [37,174]. The inhibition of ROCK also reduced proinflammatory and oxidative stress responses and improved the mitochondrial function in the liver of endotoxemic mice [175]. In contrast, the activity of Rac1 in hepatocytes was suggested as a pathway that continuously modulates inflammatory and immune responses, with both local and systemic repercussions [176].

Acute kidney injury is one of the most common clinical findings in sepsis (for review, see [177]). As previously described, the exacerbated activation of the RhoA/ROCK pathway can be involved in the maintenance of vascular responses to vasoconstrictors in the kidneys of endotoxemic rats [109]. However, studies also indicate that the modulation of Rho proteins can be essential for renal function in sepsis. For instance, evidence of an augmented activity of ROCK and protective effect of ROCK inhibitors against LPS-induced renal failure was found in mice and further associated with a proinflammatory modulation evoked by the RhoA/ROCK pathway on NF-κB activation [178]. Moreover, it was described that inflammatory responses of renal endothelial cells to TNF include caspase-dependent cytoskeletal changes accompanied by activation of RhoA [179]. Interestingly, the inhibition of ROCK also avoided the augmented permeability induced by TNF in mouse renal endothelial cells and human glomerular endothelial cells [180]. Both LPS- and adriamycin-induced podocyte cytoskeleton disruption and apoptosis were associated with reduced activity and diminished expression levels of RhoA, reinforcing the putative contribution of Rho proteins for the development of renal dysfunction in sepsis [181]. Indeed, multiple Rho proteins can be involved in this process, since both LPS and NO increased the podocyte permeability to albumin in a Cdc42/Rac1-dependent manner, and the genetic depletion of Rac1 in the bone marrow-derived macrophages protect mice against the renal injury induced by LPS administration [182].

## 7. Final Remarks and Conclusions

Although many efforts have been made in basic research to prove how Rho proteins and downstream pathways contribute to the development of sepsis and septic shock, it is likely that the lack of system selective agents, perhaps acting as activators or inhibitors on distinct Rho proteins and downstream targets, is the key missing point, preventing this pathway from being adequately explored in the clinical management of sepsis. In fact, studies in this area have been overly focused on RhoA, while other small GTPases such as RhoB and RhoC, which also have regulatory effects on cardiovascular biology and may share downstream targets such as ROCK (i.e., [183,184,185,186]), remain scarcely explored in sepsis-associated experimental models. For instance, RhoB was found to be increased in macrophages, lung, liver, and kidney of LPS-treated mice, and the knockdown of this protein prevented the transcriptional activity of NF-κB [187].

The impaired activity of Rho-associated pathways, mainly increased activity of RhoA and the inhibition of Rac1, are listed in several studies associated with experimental sepsis models. Notably, the modulation of Rho signaling, such as with ROCK inhibitors, improved the function of, or prevented damage in multiple organs, including the lungs, liver, kidneys, and brain of animals subjected to septic insults, as illustrated in Figure 1. Although some differences existed when distinct endothelial cell lines or experimental conditions were evaluated, most experimental data indicate that Rho proteins can either be modulated by proinflammatory cascades or modulators of the inflammatory response, contributing to the endothelial barrier dysfunction existing in sepsis (Figure 2).

Although ROCK has been considered the main target of RhoA, this small GTPase can interact and modulate the activity of different intracellular components, including transcription factors, the mammalian Diaphanous homolog 1 (mDia) and profilin-1 signaling pathway, and the type I phosphatidylinositol 4-phosphate 5-kinase isoforms, among others [188,189,190]. Nonetheless, the pharmacological modulation of Rho proteins has been limited to few agents that are able to inhibit only the RhoA/ROCK pathway, mainly acting on ROCK. Moreover, drugs with direct effects on Rho proteins, including but not limited to RhoA, Rac1, and Cdc42, remain to be further explored in sepsis. The association between the spread distribution, multiple roles, and few pharmacological agents has hindered the use of Rho proteins and downstream pathways as therapeutic targets. Despite the accumulated knowledge and previous clinical trials in different fields, including the cardiovascular system, the only marketed drug in the United States containing a ROCK inhibitor, named netarsudil, is an ophthalmic solution indicated for glaucoma [191]. Ripasudil has similar uses in Japan [192]. Fasudil, one of the ROCK inhibitors used in several studies included in this review, has also been used in Japan as an injectable formulation to reduce cerebral vasospasm following subarachnoid hemorrhage [193] but is still unapproved by other governmental regulatory healthy agencies. ROCK inhibitors, however, remain in focus for different diseases, including some intractable ones, such as amyotrophic lateral sclerosis [194,195]. Importantly, there are two isoforms of ROCK, named ROCK-I and ROCK-II, which are widely expressed [196] and are not distinguished by the currently available ROCK inhibitors. Moreover, although the MYPT-1 subunit of MLCP often appears as the only ROCK target explored in studies involving sepsis, ROCK has several other downstream targets [197] that may be druggable and remain to be explored.

Unlike endothelial cells, in the vascular smooth muscle cells, where the RhoA/ROCK pathway acts as a calcium sensitization mechanism mediating contractile responses, most of the studies indicate that the system is depressed (Table 2). The reduced activity of Rho components decreases the inhibitory action of ROCK on myosin phosphatase, contributing to vasodilation, vasoplegia, and the hypotensive state in septic shock (Figure 3). Thus, considering the studies that explored the relationship between vascular reactivity and Rho proteins in sepsis models, the activity of the RhoA/ROCK pathway should be enhanced to increase the vascular tone during the septic insult. However, to improve endothelial function, the desired effect appears to be the inhibition of RhoA and the activation of Rac1. Together, these data clearly indicate the existence of tissue-specific differences in how sepsis-mediated events modulate the activity of Rho proteins and correlated downstream pathways. Thus, it appears that the development of strategies to adequately modulate the activity of Rho proteins and their downstream pathways during the septic insult depends on the development of innovative tools, such as drugs able to activate or inhibit Rho proteins and related pathways selectively, refinements in mRNA-based therapeutics, and continuous advances to improve our understanding about how the system works in diseased systems.

Indeed, as detailed in this review, the majority of the studies presented in the literature used only in vitro approaches, mainly cells exposed to punctual stimuli, such as LPS, proinflammatory cytokines (TNF-α, IL-1β), or bacterial components (CNF1), creating an experimental environment whose results need to be carefully evaluated and explored in more complex models before being transposed to the septic condition. Moreover, the conclusions taken from those studies that included experiments in animals were limited by:(i)Drugs with pleiotropic or indirect effects, and unknown molecular mechanisms regarding the effects on Rho proteins and downstream targets (i.e., statins);(ii)A lack of information regarding which Rho protein is affected in different organs during the ongoing sepsis;(iii)A single point of evaluation;(iv)The usage of LPS instead of more reliable experimental models of sepsis (i.e., CLP);(iv)A lack of dose–response evaluation;(vi)The use of acute treatments only, often as a pretreatment, missing details regarding the benefices of a post-treatment and continuous therapy, and;(vii)The absence of toxicological and safety evaluation.

In summary, the published data showing the involvement of small GTPases in experimental models potentially associated with sepsis indicate that Rho proteins and their downstream pathways deserve further consideration as a valuable target for sepsis treatment.

## Figures and Tables

**Figure 1 cells-10-01844-f001:**
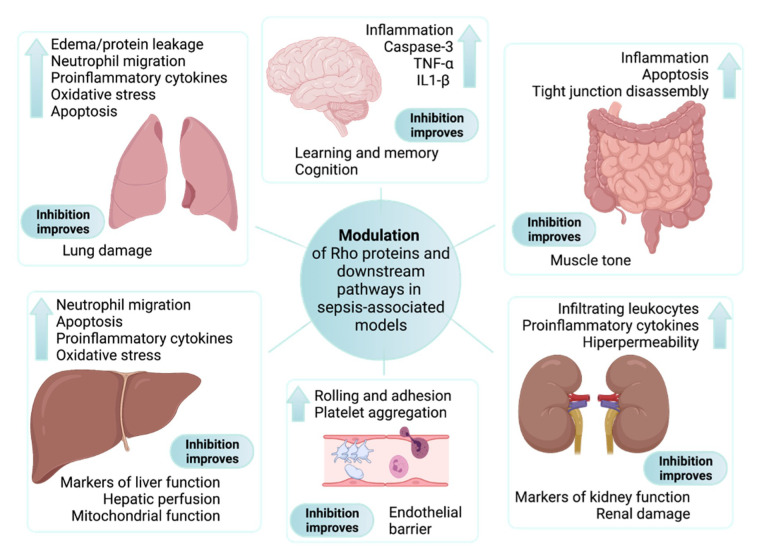
Main findings associated with the modulation of Rho proteins and downstream pathways in sepsis-associated experimental models. Although different Rho proteins have been explored in in vitro experimental approaches, all parameters presented inside each box were described in studies performed on animals subjected to sepsis models and treated with selective inhibitors of ROCK. Findings associated with sepsis-induced organ dysfunction are shown next to the up arrow. The main beneficial effects ascribed to inhibition of the RhoA/Rho-kinase pathway are included at the bottom of each box.

**Figure 2 cells-10-01844-f002:**
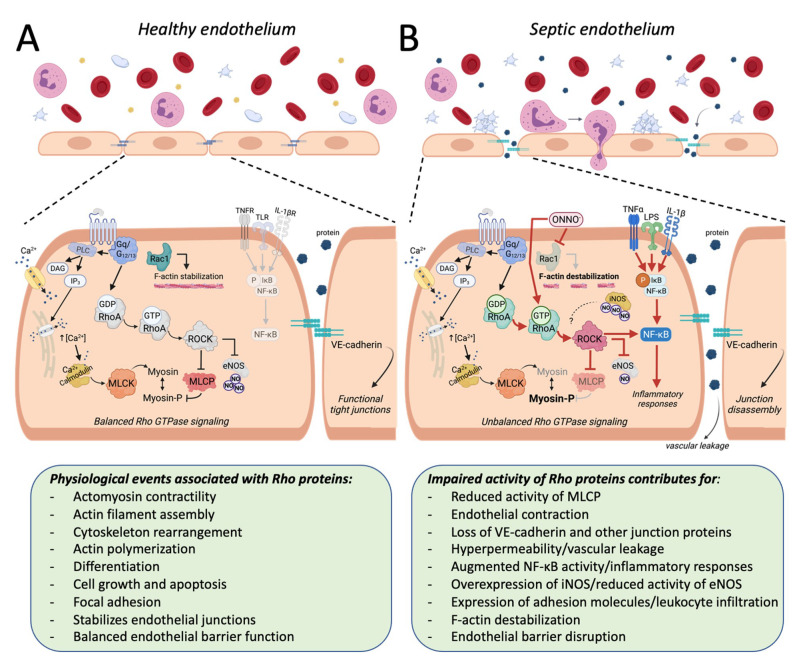
The influence of Rho proteins on the endothelial function in sepsis-associated experimental models. Rho proteins have been involved in several physiological responses. In healthy endothelial cells (**A**), these responses include Rac1-mediated F-actin stabilization and the regulatory effect of the RhoA/ROCK pathway on the activity of enzymes such as the myosin light chain phosphatase (MLCP) and the endothelial nitric oxide synthase (eNOS). The balanced activity of Rho proteins contributes to the maintenance of the endothelial barrier function. Under experimental models of sepsis (**B**), systemic or locally produced mediators (i.e., ONNO, cytokines) reduce the activity of Rac1 (light gray arrow), contributing to F-actin destabilization, and the activity of the RhoA/ROCK pathway is increased (see the enhanced colors and ticker red arrows), resulting in reduced inhibition of MLCP and augmented levels of phosphorylated myosin (Myosin-P). This unbalanced Rho signaling contributes to exacerbating inflammatory responses, endothelial contraction, loss of junction proteins (i.e., VE-cadherin), endothelial barrier dysfunction, leukocyte infiltration, and vascular leakage, among others. The question mark coming from iNOS in panel B indicates that it remains unclear whether the high amounts of NO produced during septic insults significantly influence the activity of the RhoA/ROCK pathway in endothelial cells. PLC, phospholipase C; DAG, diacylglycerol; IP3, phosphatidylinositol 1,4,5-trisphosphate; MLCK, myosin light chain kinase; TNFR, tumor necrosis factor α receptor; TLR, Toll-like receptor 4; IL-1βR, interleukin 1β receptor.

**Figure 3 cells-10-01844-f003:**
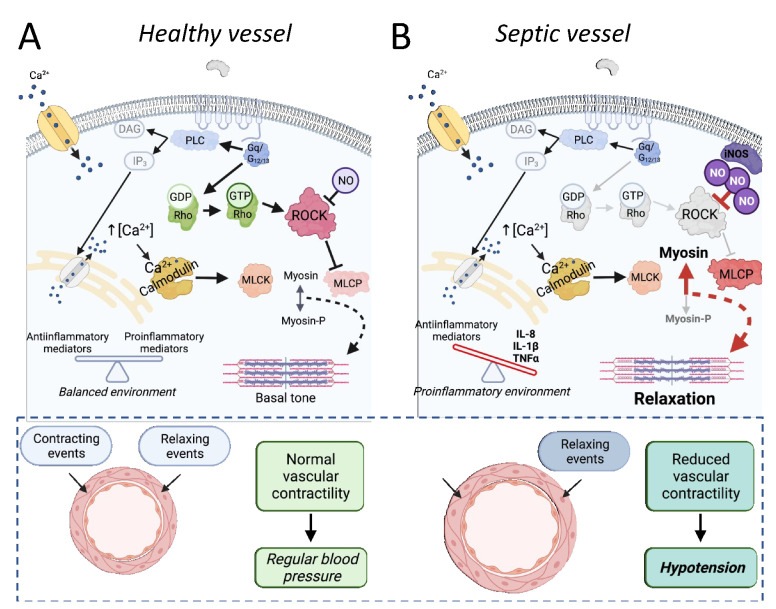
The involvement of Rho proteins in sepsis-associated reduced vascular dysfunction. In healthy vessels (**A**), activation of the RhoA/ROCK pathway functions as a calcium sensitization mechanism, crucial for regulating vascular tone. Once activated by RhoA, ROCK inhibits the myosin light chain phosphatase (MLCP) activity, maintaining balanced levels of phosphorylated myosin and the vascular tone. Under physiological conditions, there is an equilibrium between contracting and relaxing events, resulting in normal vascular contractility and normal blood pressure. In septic vessels (**B**), the proinflammatory environment and high levels of nitric oxide, among other factors, reduce the effectivity of Rho-mediated calcium sensitization, as illustrated by the grayscale of RhoA/ROCK pathway components and arrows. The final result is an augmented MLCP activity and less phosphorylated myosin (Myosin-P), contributing to the loss of vascular tone and relaxation of vessels. Under septic conditions, the relaxing events are enhanced, and the reduced vascular contractility in resistance arteries is one of the main causes of hypotension. Importantly, the absence of changes in other components included in the scheme (i.e., membrane receptors, calcium levels) does not mean that they are not compromised in sepsis. PLC, phospholipase C; DAG, diacylglycerol; IP3, phosphatidylinositol 1,4,5-trisphosphate; MLCK, myosin light chain kinase.

**Table 1 cells-10-01844-t001:** The involvement of Rho proteins and downstream pathways in the endothelial function as found in sepsis-associated experimental models.

System and Model	Component(s) Evaluated	Experimental Setup	Impact on the System and Main Findings ^a^	Ref.
HLMVEC; LPS	Rac1, RhoA	Indirect modulation + siRNA (includes in vivo evaluation)	Activation of Rac1 and inhibition of RhoA prevents vascular leakage; stabilizes VE-cadherin.	[43]
	ROCK	Direct inhibition + siRNA	ROCK can prevent or enhance vascular leakage.	[44]
	RhoA	Direct inhibition + siRNA (includes in vivo evaluation)	Inhibition of RhoA nitration prevents vascular leakage.	[72]
	RhoA, ROCK	Indirect modulation + direct inhibition	Inhibition of RhoA/ROCK prevents vascular leakage.	[51]
	Rac1	Direct modulation (includes in vivo evaluation)	Prevention of Rac1 nitration limits vascular leakage.	[73]
HLMVEC; LTA + PepG	MCLP, MLC	Direct inhibition (includes in vivo evaluation)	Inhibition of ROCK prevents vascular leakage.	[53]
HLMVEC; TNF-α	RhoA, ROCK	Direct inhibition	Inhibition of RhoA/ROCK pathway prevents vascular leakage.	[55]
HUVEC; LPS	MLC, pMLC	Direct inhibition	Inhibition of ROCK prevents vascular leakage.	[31]
	RhoA, GEF-H1	iRNA	Reduction in RhoA activation disfavor inflammatory pathway.	[62]
	RhoA, MLCP	siRNA	RhoA inhibition prevents vascular leakage and stabilizes VE-cadherin.	[39]
	ROCK, RhoA, MLCP	Direct inhibition (includes in vivo evaluation)	Inhibition of ROCK prevents vascular leakage.	[66]
	ROCK, GEF-H1	Direct inhibition + siRNA	Inhibition of ROCK and GEF-H1 prevents vascular leakage, stabilizes adherens and tight junctions.	[89]
	RhoA, Rac, Cdc42	Indirect modulation + siRNA	Inhibition of RhoA and Rac prevents vascular leakage, stabilizes junctions, and disfavors inflammation.	[92]
	RhoA	Direct inhibition	Inhibition of ROCK prevents vascular leakage and stabilizes VE-cadherin.	[40]
	ROCK	Indirect modulation + direct inhibition	Downregulation of ROCK disfavor inflammatory pathway.	[90]
	RhoA, ROCK	Direct inhibition + siRNA	Inhibition of ROCK reduces stress fiber formation.	[91]
HUVEC; TNF-α	RhoA, ROCK, MLCP	Indirect modulation	Inhibition of ROCK prevents vascular leakage and stabilizes VE-cadherin.	[56]
HUVEC; HBP	ROCK	Indirect modulation + direct inhibition	Inhibition of ROCK prevents vascular leakage.	[50]
HVEC; thrombin	RhoA, ROCK	Indirect modulation + siRNA	Inhibition of ROCK prevents vascular leakage, stabilizes VE-cadherin, reduces stress fiber formation.	[49]
HPMEC;	ROCK	Direct inhibition	Inhibition of ROCK disfavor inflammatory and coagulation pathways.	[64]
LPS	ROCK	Direct inhibition	Inhibition of ROCK reduces vascular leakage and apoptosis.	[35]
	Rho-GTP	Indirect modulation	Downregulation of ROCK prevents vascular leakage, stabilizes protein junctions (i.e., VE-cadherin).	[94]
HPAEC; LPS	Rac, Cdc42, MLC	Indirect modulation	Rac1 and Cdc42 activation prevents vascular leakage, stabilizes VE-cadherin.	[93]
HPAEC; IL-6	ROCK	Direct inhibition + siRNA (includes in vivo evaluation)	Inhibition of ROCK prevents vascular leakage, stabilizes VE-cadherin, avoids leucocyte adhesion.	[58]
HPAEC; *Sthaphyloccocus aureus*	Rho-GEF-H1	siRNA	Inhibition of GEF-H1 prevents vascular leakage, disfavor inflammatory pathway.	[63]
HDMEC; LPS	RhoA, Rac1	Direct inhibition	Downregulation of Rac1 worse vascular leakage.	[57]
HDMEC;	RhoA	Indirect modulation	Inhibition of RhoA reduces leukocyte migration.	[46]
TNF-α	ROCK, MLC	Direct inhibition + siRNA	Inhibition of ROCK prevents vascular leakage.	[59]
HDMEC; CNF-1	RhoA, Rac1, Cdc42	Direct inhibition or activation	ROCK enhances and Rac1/Cdc42 reduce vascular leakage; Rac1/Cdc42 inactivation worse junction stability.	[52]
MPVEC; LPS	ROCK	miRNA	Inhibition of ROCK prevents apoptosis and inflammation.	[95]
PAEC; CNF-1	RhoA, Rac1, Cdc42, ROCK	Direct inhibition or activation	ROCK enhances and Rac1/Cdc42 reduce vascular leakage; Rac1/Cdc42 inactivation worse junction stability.	[52]
	ROCK, pMLC	RNAi (includes in vivo evaluation)	Inhibition of ROCK prevents vascular leakage, stabilizes connexin 43.	[33]
MyEnd; CNF-1	Rac1, Cdc42	Indirect modulation (includes in vivo evaluation)	Upregulation of Rac1 and Cdc42 improves the endothelial barrier.	[48]
	RhoA, Rac1, Cdc42, ROCK	Direct inhibition or activation	ROCK enhances and Rac1/Cdc42 reduce vascular leakage; Rac1/Cdc42 inactivation worse junction stability.	[52]
MesEnd; CNF-1	RhoA, Rac1, Cdc42, ROCK	Direct inhibition or activation	ROCK enhances and Rac1/Cdc42 reduce vascular leakage; Rac1/Cdc42 inactivation worse junction stability.	[52]
bEnd.3; LPS	RhoA; GEF	Direct inhibition + siRNA	Inhibition of RhoA and GEF prevents vascular leakage, stabilizes zonnula occludent 1 and reduces stress fiber formation.	[88]
Pericytes; LPS	RhoA	Indirect modulation	Activation of RhoA prevents vascular leakage.	[45]
LSEC; LPS	ROCK	Direct inhibition	Inhibition of ROCK and its nitration prevents vascular leakage.	[71]
Mice lung; LPS	Rho-GTP, ROCK, MLCP	Indirect modulation	Downregulation of Rho-GTP, ROCK, and MLCP prevents vascular leakage.	[34]
Rat mesenteric artery; LPS	ROCK	Direct inhibition	Inhibition of ROCK prevents vascular leakage and avoids leucocyte adhesion.	[67]
Guinea pig skin; LPS	ROCK	Direct inhibition	Inhibition of ROCK prevents vascular leakage.	[36]

^a^: only those conclusions directly associated with Rho proteins were included. Abbreviations: bEnd.3, mouse brain endothelial cells; CLP, cecal ligation and puncture; CNF-1, *Escherichia coli* cytotoxic necrotizing factor 1; HBP, heparin-binding protein; HDMEC, human dermal microvascular endothelial cells; HLMVEC, human lung microvascular endothelial cells; HPAEC, human pulmonary artery endothelial cells; HPMEC, human pulmonary microvascular endothelial cells; HUVEC, human umbilical vascular endothelial cells; LSEC, liver sinusoidal endothelial cells; LPS, lipopolysaccharide; LTA, lipoteichoic acid; MesEnd, microvascular mesenteric endothelial cells; MLC, myosin light chain; MCLP, myosin light chain phosphatase; MPVEC, murine pulmonary microvascular endothelial cells; MyEnd, mouse myocardial endothelial cells; PAEC, porcine aorta endothelial cells; PAF: platelet-activating factor; PepG: *Staphylococcus* aureus-derived peptidoglycan; pMLC, phosphorylated myosin light chain.

**Table 2 cells-10-01844-t002:** The involvement of Rho proteins and downstream pathways in the cardiovascular function as found in sepsis-associated experimental models.

System and Model	Component(s) Evaluated	Experimental Setup	Impact on the System and Main Findings ^a^	Ref.
SM artery; Rats; LPS ^b^	ROCK	Direct inhibition + functional + molecular approaches	Upregulation of ROCK enhances contractile responses.	[98]
	RhoA, ROCK	Indirect modulation + direct inhibition + functional + molecular approaches	RhoA is reduced; activation of the pathway improves contractile responses.	[110]
SM artery; Rabbits; IL-1β ^c^/TNF-α ^c^	ROCK, MLCP	Direct inhibition + functional + molecular approaches	Inhibition of ROCK contributes to IL-1β-induced vascular hyporeactivity.	[103]
	ROCK, MLCP	Functional + molecular approaches	Downregulation of ROCK contributes to TNF-α-induced vascular hyporeactivity.	[104]
RM artery; Rats; LPS^b^	RhoA, ROCK, MLCP	Direct inhibition + functional + molecular approaches	Upregulation of Rho components fails to trigger contractile responses; RhoA/ROCK is inhibited by the nitric oxide/guanylate cyclase pathway.	[101]
Aorta; Rats; LPS ^b^	MLC, ROCK	Functional + molecular approaches	Hyporeactivity to ET-1 does not involve the RhoA/ROCK pathway.	[100]
	RhoA, ROCK, MLCP	Direct inhibition + functional + molecular approaches	The activity of RhoA increases increases (1–2 h) and reduces (4–6 h) after LPS. Norepinephrine-induced vasoconstriction is more sensitive to ROCK inhibition.	[102]
Aorta; Rats; CLP	RhoA, ROCK, MLCP	Direct inhibition + functional + molecular approaches	Upregulation of RhoA and ROCK at 60 days after CLP; augmented activation of RhoA/ROCK pathway accounts for enhanced contractile responses to angiotensin II.	[108]
Renal vascular bed and blood pressure; Rats; LPS ^b^	RhoA, ROCK, MLCP	Direct inhibition + functional + molecular approaches (includes in vivo treatment/evaluation)	Increased RhoA/ROCK in the renal vascular bed accounts for enhanced pressor responses to vasopressin.	[109]
Femoral artery; Mice; CLP	MLCP	Functional + molecular approaches	Thromboxane A2-induced vasoconstriction and phosphorylation of MLCP were reduced 5 days after CLP.	[107]
Pulmonary artery; Rats; LPS ^c^	RhoA	Direct activation + molecular approaches	RhoA activation prevents vascular damage/F-actin rearrangement.	[96]
VSMC; Rats; LPS/IL-1β ^c^	RhoA, ROCK, Rac1, MLCP	Direct inhibition + molecular approaches	LPS reduces RhoA activity. IL-1β increases RhoA activity. ROCK negatively modulates NF-κB.	[97]
Blood pressure; Rats; CLP	RhoA, ROCK	Direct inhibition + systemic effects + molecular approaches (includes in vivo treatment/evaluation)	RhoA/ROCK pathway is up-regulated; inhibition of ROCK improves blood pressure.	[35]
Cardiomyocytes; Rats; LPS ^c^	RhoA	Molecular approaches	RhoA expression and activity are further increased by LPS and nitric oxide in tissues from diabetic animals.	[111]
Heart; Mice; LPS ^b^	Rac1	Molecular approaches	Lack of Rac1 reduces inflammatory markers, including TNF.	[112]
	ROCK	Direct inhibition + functional + molecular approaches (includes in vivo treatment)	Inhibition of ROCK improved contractile function and mitochondrial respiration.	[113]
Cardiomyocytes and heart; Mice; LPS ^b,c^	Rac1	Functional + molecular approaches	Rac1 expression and activity are increased; lack of Rac1 reduces TNF and improves cardiac function.	[114]
H9C2; TNF-α ^c^	RhoA, Cdc42, Rac1	Direct inhibition + molecular approaches	TNF-α increases RhoA, and ROCK inhibition attenuates mitochondrial fragmentation.	[115]
Heart and H9C2; Rats; LPS	RhoA, ROCK	Indirect modulation + molecular approaches (includes in vivo treatment)	RhoA/ROCK proteins are increased, and the reduction is associated with improved cardiac function and reduced apoptosis.	[116]
Papillary muscle; Rats; CLP	ROCK	Indirect modulation + functional + molecular approaches (includes in vivo treatment)	Block of ROCK avoids intermedin 1-53-mediated cardiac troponin I phosphorylation.	[117]

^a^: only those conclusions directly associated with Rho proteins were included. ^b^: LPS was administered in vivo (the time for in vitro evaluation varied between studies). ^c^: LPS, IL-1β or TNF-α were incubated in vitro. VSMC, vascular smooth muscle cells. Abbreviations: CLP, cecal ligation and puncture; H9C2, rat cardiomyoblast cells; LPS, lipopolysaccharide; MLC, myosin light chain; MCLP, myosin light chain phosphatase; RM, resistance mesenteric (artery); SM, superior mesenteric (artery).

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
