# Peer review of "Rho-Proteins and Downstream Pathways as Potential Targets in Sepsis and Septic Shock: What Have We Learned from Basic Research"

_cells, 2021, doi:10.3390/cells10081844_

Round 1

Reviewer 1 Report

Sepsis is a complex disease involving many different organs and cell types that dynamically changes during progression from an initial “cytokine storm” to an immunosuppression phase. Rho GTPases, on the other hand, are a family 20 different molecules with a huge number of different effectors, often ubiquitously expressed, regulated mostly on difficult to test activation level and cell type specific effects based on the set of effector molecules expressed. Trying to give a review on Rho GTPases function in sepsis with clear take-home messages is therefore extremely difficult due to the complexity of both issues.

The authors tried, but in the current form it is difficult for the reader to obtain clear insights, both with respect to what is known for sure and what are the most important open questions.

Specific comments:

  1. The review could benefit from a focus on RhoA, as the majority of the data are obtained with ROCK inhibitors. However, the closely related RhoB and RhoC should mentioned (which also activate ROCK) and the distinction of ROCK1 and ROCK2. Other Rho GTPases effectors than ROCK should also be mentioned to make the reader aware that ROCK is only one of many effectors of RhoA/B/C. ROCK inhibitors inhibit therefore only one aspect of Rho signaling.
  2. A critical evaluation of the methodology of investigating Rho GTPase function is required.

- How specific are the drugs used? Statins fx are changing the metabolism and reducing prenylation of all prenylated molecules not specifically of Rho GTPases. Assumed it would be only Rho GTPases, would the results be interpreted as a combined effect on all prenylated Rho GTPases simultaneously? What is the extent of Rho GTPase inhibition obtained? CNF1 is reported to activate Rho, Rac Cdc42. How can CNF1 results be interpreted?

- How specific are the ROCK inhibitors?

- How can the effects of systemic drugs be interpreted in an multiorgan disesase with an immune system running wild? What are the target cells? What are direct effects? What are indirect effects?

- How relevant are single cell in vitro systems treated with a single molecule such as LPS or TNF for understanding the situation during sepsis?

- Have tissue-specific ko of Rho GTPases or Rho GTPase effectors been tested in sepsis models? Would this be something that should be tested more in the future?

  1. Next it would be nice if the authors could stress more which Rho GTPase effect in a specific organ in sepsis is relatively clear and what are the open questions.
  2. Rho/ROCK increases cell contraction contributing to vascular leakage of endothelial cells in sepsis. Smooth muscle cells, however, relax in sepsis, correlating with decreased Rho/ROCK signaling. This tissue specific differences in regulation could be highlighted more.
  3. Figures should be improved:

-Fig. 1 is not giving any clear idea of which Rho GTPase is doing what in which organ

- Fig. 2 should show how Rho ROCK myosin phosphorylation together with actin stress fibers induced by RhoA contribute to contraction. Should iNOS increase not be shown?

- Fig. 3 gives the impression that NO inhibits ROCK directly, which raises the questions why it is not doing this in the adjacent endothelial cells.

Author Response

RESPONSES TO COMMENTS FROM REVIEWER #1:

(For convenience, a pdf file containing the same responses was also attached) 

General comments:

“Sepsis is a complex disease involving many different organs and cell types that dynamically changes during progression from an initial “cytokine storm” to an immunosuppression phase. Rho GTPases, on the other hand, are a family 20 different molecules with a huge number of different effectors, often ubiquitously expressed, regulated mostly on difficult to test activation level and cell type specific effects based on the set of effector molecules expressed. Trying to give a review on Rho GTPases function in sepsis with clear take-home messages is therefore extremely difficult due to the complexity of both issues.

The authors tried, but in the current form it is difficult for the reader to obtain clear insights, both with respect to what is known for sure and what are the most important open questions.”

Response to Reviewer #1 – General comments: We thank you very much for your time and attention with the data presented in this review. All points raised were carefully addressed and replied. We fully agree with all comments provided by the reviewer, and we hope that the way we treated each one can allow the reader to get the same insights and feelings that we had while writing and revising this document. As requested by the editor, all changes (except for adjustments in tables and figures) remain marked with the review tools from Microsoft Word. Once again, thanks for sharing your opinion and expertise in the field. 

Specific comments:

Comment #1: “The review could benefit from a focus on RhoA, as the majority of the data are obtained with ROCK inhibitors. However, the closely related RhoB and RhoC should mentioned (which also activate ROCK) and the distinction of ROCK1 and ROCK2. Other Rho GTPases effectors than ROCK should also be mentioned to make the reader aware that ROCK is only one of many effectors of RhoA/B/C. ROCK inhibitors inhibit therefore only one aspect of Rho signaling.” 

Response to reviewer #1 – Comment #1: We also felt that RhoA could be the only Rho protein explored in this review. However, that would be the only one just because other members were not extensively explored in the experimental conditions required to be included by us as a sepsis-associated model. The studies including Rac1 and Cdc42, for instance, would be suppressed, a choice that would not help the reader to get inside what comes (or is missing) from basic research. We were delighted with this comment because it gave us the correct stimulus to state what is missing, not only for Rho family members but also for targets. The following changes, performed in our submission, were done to address this comment: i) page 15, lines 574-579.

Comment #2: “A critical evaluation of the methodology of investigating Rho GTPase function is required.”  

Response to reviewer #1 – Comment #2:  The points were listed from 2a-2e and are replied below.

Comment #2a: “How specific are the drugs used? Statins fx are changing the metabolism and reducing prenylation of all prenylated molecules not specifically of Rho GTPases. Assumed it would be only Rho GTPases, would the results be interpreted as a combined effect on all prenylated Rho GTPases simultaneously? What is the extent of Rho GTPase inhibition obtained? CNF1 is reported to activate Rho, Rac Cdc42. How can CNF1 results be interpreted?”  

Response to reviewer #1 – Comment #2a:  The reviewer is correct, and concerning the intracellular pathways explored, these drugs do not have molecular mechanisms described. Our interpretation for all these studies was that some indirect modulation was being achieved, independently of conclusions claimed by authors. To bypass this limitation without add additional unrelated information in our review, maintaining the citation of these studies (since they can be found relevant for the field), we added a new item reinforcing the unknown mechanism and pleiotropic effects that may be involved. Additional targets for RhoA and their relevance in this topic were also included in this revised version. The following changes, performed in our submission, were done to address this comment: i) page 17, lines 638-639; ii) page 16, lines 590-594; iii) page 16, lines 610-613 (which also added other ROCK targets).

Comment #2b: “How specific are the ROCK inhibitors?”  

Response to reviewer #1 – Comment #2b:  The classical ROCK inhibitors (i.e., Y compound) present a reasonable selectivity for ROCK-I and ROCK-II (over 1000x) compared to other classical kinases. Of course, such studies were performed in vitro, under controlled and optimized conditions. Thus, as for any drug, there is no warranty of selectivity when other unrelated targets are considered.  We found this question very important, mainly thinking about the two ROCK isoforms, which have not been adequately explored as separate enzymes in different systems, neither in several physiological nor diseased conditions. The following changes, performed in our submission, were done to address this comment: i) page 16, lines 608-610.

Comment #2c: “How can the effects of systemic drugs be interpreted in an multiorgan disesase with an immune system running wild? What are the target cells? What are direct effects? What are indirect effects?”  

Response to reviewer #1 – Comment #2c:  That is a good question for which no proper response can be detailed. This lack of information was emphasized in the list of limitations associated with in vivo studies explored in our review.  The following changes, performed in our submission, were done to address this comment: i) page 7, lines 640-641.

Comment #2d: “How relevant are single cell in vitro systems treated with a single molecule such as LPS or TNF for understanding the situation during sepsis?”  

Response to reviewer #1 – Comment #2d:  We believe such studies are relevant as experimental models potentially applied in the field. It includes both the use of cultured cells, as well as LPS or any other isolated stimulus. We did agree with the reviewer that this point could be emphasized in our manuscript. The following changes, performed in our submission, were done to address this comment: i) page 16, lines 632-635.

Comment #2e: “Have tissue-specific ko of Rho GTPases or Rho GTPase effectors been tested in sepsis models? Would this be something that should be tested more in the future?”  

Response to reviewer #1 – Comment #2e:  We did not find any information about that. However, we fully agree with the reviewer's insight. Once the tissue-specific ko can be appropriately obtained, it would be of great value to understand the biological roles of the missing Rho GTPases (or effectors). However, thinking about future therapeutic perspectives, further advances in innovative methods, such as mRNA-based approaches, could be "easily" translated for the clinic (see page 16, lines 625-630; it includes our original considerations about that).

Comment #3: “Next it would be nice if the authors could stress more which Rho GTPase effect in a specific organ in sepsis is relatively clear and what are the open questions.”

Response to reviewer #1 – Comment #3: We did our best to state the role of Rho proteins without feeling it was too speculative. The following changes, performed in our submission, help us address this comment: i) figure 1 was improved, and the legend detailed; ii) page 16, lines 623-625; iii) page 17, lines 640-641.

Comment #4: Rho/ROCK increases cell contraction contributing to vascular leakage of endothelial cells in sepsis. Smooth muscle cells, however, relax in sepsis, correlating with decreased Rho/ROCK signaling. This tissue specific differences in regulation could be highlighted mor..”

Response to reviewer #1 – Comment #4: We agree with the reviewer, and this information was also emphasized in our final remarks (page 46, lines 623-625).

Comment #5a: Fig. 1 is not giving any clear idea of which Rho GTPase is doing what in which organ”

Response to reviewer #1 – Comment #5a: Thanks for this clear and vital criticism since the main aim of any figure must be a clear message. We did reorganize the figure and its legend to follow your message, and we believe that the new version of Figure 1 is much better than the previous. This figure aimed to be introductory, adding value for studies performed in vivo – which were subsequently detailed. All pharmacological treatments consisted of ROCK inhibitors, although the authors had occasionally addressed expression levels of other GTPases. We hope the figure and the revised legend can now give a better idea of this information. As previously mentioned, the current literature does not allow us to trace a clear profile of tissue-specific Rho-GTPases involved in the beneficial or deleterious responses to the septic insult.

Comment #5b: Fig. 2 should show how Rho ROCK myosin phosphorylation together with actin stress fibers induced by RhoA contribute to contraction. Should iNOS increase not be shown?”

Response to reviewer #1 – Comment #5b:  Although we agree with the first part of this comment (I mean, it would be excellent to include additional components, such as cofilin, LIM kinase, and others), we had no space for that. As the reviewer will not, the central part of the Figure was revised to improve its legibility. We fully agree with the second part of this comment, and we added iNOS inside the endothelial cell in the septic cell (Figure 2B, central part). However, it is essential to state that, although the effect of NO on RhoA/ROCK has been suggested in vascular smooth muscle cells, the same is not valid for endothelial cells. We added a line with a question inside the Figure and explained it in the legend (now on page 8, lines 288-290).

Comment #5c: Fig. 3 gives the impression that NO inhibits ROCK directly, which raises the questions why it is not doing this in the adjacent endothelial cells.”

Response to reviewer #1 – Comment #5c: Yes, the reviewer is right, and I believe that the changes performed in Figure 2 did address this rational, without be speculative (please check our previous response; comment listed as #5b).

Reviewer 2 Report

This review on Rho signaling in sepsis and septic shock addresses a very timely topic. It discusses the participation mainly of RhoA in this pathologic conditions with an emphasis on the cardiovascular system, which is by far the most extensively studied target, and provided some information about potential therapeutic interventions. The review is very comprehensive and the authors have made a great effort at synthesizing recent literature on the topic. The manuscript is in principle logically organised, there are however a few aspects that would require some rearrangement to improve the flow of the paper. I’m very positive about this review but I would like to ask the authors to address following issues

  1. The introduction should open with the second paragraph (line 41). The first paragraph should go at the end of the introduction and the first sentence is unnecessary. “We aim to provide” rather than “the authors aim to provide…”
  2. The sentences in lines 88-91 are misleading. They claim Rho proteins are activated by GPCRs in a similar way as heterotrimeric G proteins, which is incorrect. Hetrotrimeric G proteins bind GPCRs directly and are also directly activated by these receptors whereas Rho (and in general small GTPases) are not. Signals from the receptors are first transmitted to GEFs. These sentences need to be modified.
  3. The sentence in lines 100-103 is repetition of information presented earlier in the introduction.
  4. The tables are very difficult to read because there is no clear delimitation of rows. A clear layout is needed.
  5. Tables 1 and 2 should be mentioned at the beginning of the respective section, where their purpose should be highlighted.
  6. I recommend splitting section 4 into two sections, one on vascular function and one on heart function, to make reading easier. There is no need to split Table 2 but it can be done too if the authors wish so.
  7. Figures 2 and 3 would make a better service within the corresponding sections (3 and 4). They should be moved and commented on there. The central part of figure 2 is too small for proper reading, it should be enlarged.
  8. The final remarks section should be reorganised. The first sentence (lines 510-511) should be deleted, it’s an obvious statement. This section should open with the second paragraph (line 525) and be continued with the first paragraph (line 511) and then the last one (line 551).
  9. Line 514: there are actually various inhibitors of Rac and Cdc42 (as well as Rho directly). The authors probably mean drugs with effects on signaling regulated by other Rho proteins (in analogy to drugs regulating RhoA signaling mentioned in the previous sentence.

Correct following typos/errors:

Greek letters are lost throughout

Nitric oxide appears both in full and abbreviated as NO. Be consistent.

Line 13, should be “and septic shock remain…”

Line 103, should be adaptive (not adaptative)

Lines 174-175, the Clostridium C3 exoenzyme inhibits and CNF1 activates Rho, therefore the sentence should read: Moreover, both activation and inactivation …… by E. coli cytotoxic  … and Clostridium C3 exoenzyme, respectively.

Line 180, Cdc42.

Line 342: Nonetheless, administration of the selective ROCK inhibitor fasudil to rats…

Line 454: something missing in this sentence. The incubation of ??? with Y-27326…

Line 506, Cdc42, rather than CDC42

Line 532: modulation of Rho signaling, such as…

Author Response

RESPONSES TO COMMENTS FROM REVIEWER #2:

(For convenience, a pdf file containing the same responses was also attached) 

“This review on Rho signaling in sepsis and septic shock addresses a very timely topic. It discusses the participation mainly of RhoA in this pathologic conditions with an emphasis on the cardiovascular system, which is by far the most extensively studied target, and provided some information about potential therapeutic interventions. The review is very comprehensive and the authors have made a great effort at synthesizing recent literature on the topic. The manuscript is in principle logically organised, there are however a few aspects that would require some rearrangement to improve the flow of the paper. I’m very positive about this review but I would like to ask the authors to address following issues”

Response to Reviewer #2 – General comments:  We thank you very much for your comments and suggestions about our work and your time and efforts to help us improve our review and its presentation of our submission. We fully agreed with all your points. As requested by the editor, all changes (except for adjustments in tables and figures) remain marked with the review tools from Microsoft Word. Once again, thanks for sharing your knowledge with us and for your efforts to improve this review.

Specific comments:

Comment #1: “The introduction should open with the second paragraph (line 41). The first paragraph should go at the end of the introduction and the first sentence is unnecessary. “We aim to provide” rather than “the authors aim to provide…”

Response to reviewer #2 – Comment #1: Indeed, this suggestion made sense and was followed. Thanks for that. The following changes, performed in our submission, were done to address this comment: i) the first sentence was deleted; ii) the information from our first paragraph is now on page 3, lines 110-115.

 Comment #2: The sentences in lines 88-91 are misleading. They claim Rho proteins are activated by GPCRs in a similar way as heterotrimeric G proteins, which is incorrect. Hetrotrimeric G proteins bind GPCRs directly and are also directly activated by these receptors whereas Rho (and in general small GTPases) are not. Signals from the receptors are first transmitted to GEFs. These sentences need to be modified.”

 Response to reviewer #2 – Comment #2: Thanks again. The sentence was revised to prevent any comprehension different from the message desired or from the current literature. The chances are on page 2, lines 94-96.

Comment #3: The sentence in lines 100-103 is repetition of information presented earlier in the introduction.” 

Response to reviewer #2 – Comment #3: It was removed as suggested (as indicated on page 3, lines 107-108).

Comment #4: “The tables are very difficult to read because there is no clear delimitation of rows. A clear layout is needed.”

Response to reviewer #2 – Comment #4: We do apologize for this problem. For sure, the tables were utterly unformatted during any step between the submission and the automatic configuration of the MS Word document (or pdf) sent for the reviewers. Both Table 1 and Table 2 were formatted again and had each line and column carefully evaluated. We hope it appears correct for all reviewers now! We will be alert for this problem in the published version of this review in case of its acceptance.

Comment #5: “Tables 1 and 2 should be mentioned at the beginning of the respective section, where their purpose should be highlighted.”

Response to reviewer #2 – Comment #5: They were mentioned, as suggested. In our submission, the following changes were made to address this comment: i) page 4, lines 166-167; ii) page 9, line 313.

Comment #6: “I recommend splitting section 4 into two sections, one on vascular function and one on heart function, to make reading easier. There is no need to split Table 2 but it can be done too if the authors wish so.”

Response to reviewer #2 – Comment #6: It was split, as suggested, and it is probably better for the reader. In our submission, the following changes were made to address this comment: i) page 4, lines 166-167; ii) page 11, line 392.

Comment #7: “Figures 2 and 3 would make a better service within the corresponding sections (3 and 4). They should be moved and commented on there. The central part of figure 2 is too small for proper reading, it should be enlarged.”

Response to reviewer #2 – Comment #7: We agree with the reviewer. The figures were moved and mentioned, as suggested. The size of Figure 2 was revised. The following changes, performed in our submission, were done to address this comment: i) page 6, lines 257-262; ii) page 9, lines 306-308.

Comment #8: “The final remarks section should be reorganised. The first sentence (lines 510-511) should be deleted, it’s an obvious statement. This section should open with the second paragraph (line 525) and be continued with the first paragraph (line 511) and then the last one (line 551).

Response to reviewer #2 – Comment #8: The section was revised following your suggestions. Additional insertions were done following suggestions from other reviewers, which we believe added value for this section.

Comment #9: “Line 514: there are actually various inhibitors of Rac and Cdc42 (as well as Rho directly). The authors probably mean drugs with effects on signaling regulated by other Rho proteins (in analogy to drugs regulating RhoA signaling mentioned in the previous sentence.”

Response to reviewer #2 – Comment #9: Thank you for this observation. We did amend the information (now in page 15, line 572).

 Comment #10: “Correct following typos/errors: […]”

Response to reviewer #2 – Comment #10: We revised all listed errors accordingly, and they appear marked along with the manuscript. We thank you very much for this detailed review and your work detailing our mistakes. We are pretty sure that such contribution indeed improved our manuscript.

Reviewer 3 Report

This manuscript from Hahmeyer & Silva-Santos is a review article discussing Rho GTPase and their downstream signaling pathways in context of sepsis and septic shock, with a special focus on the potential targets for therapeutic interventions This is clearly a great choice for a review article, as the topic lies at the interface of conserved cellular mechanisms of Rho GTPase signaling and pathophysiological relevance in sepsis. I laud the author’s effort to identify not only this critical area of research, but more importantly highlight the major ideas in the field and the confusion and contradictions in the field literature, which have limited therapeutic advancement. To do so, authors have focussed on the studies conducted in experimental models and basic research and have presented a very thorough and well referenced overview of the field. I have a few concerns which I hope the authors will consider addressing.

Concerns (in order of appearance in text):

  1. I like the framing of the title, but feel the message could be conveyed better if authors reframe the last portion into : ‘: What does basic research tell?’ or ‘:What have we learned from basic research?”

  1. Please consider switching the order of paragraph 2 (Sepsis is defined…) and paragraph 1 (this review does…). Starting with paragraph 2 will give the article the big picture view of Sepsis as a critical life-threatening condition, followed by paragraph 1 which nicely defines the scope of the review.

  1. The schematic in Figure 1 is really well illustrated but authors may want to add a little bit more explanation on the what the arrows define, and the boxes indicate. It’s could be a bit hard to follow the concept and message of the schematic. Please explain this figure in more details in the figure captions.

  1. This manuscript does a great job in field literature survey and is a very well-referenced text. However, in the introduction, line 78 onwards, where authors list out the regulatory roles of Rho proteins in different processes ranging from adhesion, polarity, motility (lines 78-81), it will be good to add one more line saying something like this “ Recent studies have also now suggested Rho-GTPase signaling pathways crosstalk with each other and are influenced by cellular mechanics, leading to self-organisation of the several dynamic cellular processes. (refer to following reviews like PMID: 29632270, 31999511, 27533896).

  1. In line 170, authors discuss inconclusive findings from ROCK inhibition on barrier function. Are there studies which look at the dose dependence of ROCK inhibition on this process, is there a switch from improvement to worsening based on the dose regime or mode of administration.

  1. Line 180, please correct to Cdc42 from 43.

  1. Figure 2, please increase the font size on the schematic to make the signaling pathway more readable.

  1. In the last section, I wonder if it’s possible to highlight the concerns with the existing studies (lines 565-570) unknown questions in a bullet point fashion. It will allow readers to carry home the distilled message of the entire review.

Author Response

RESPONSES TO COMMENTS FROM REVIEWER #3:

(For convenience, a pdf file containing the same responses was also attached) 

This manuscript from Hahmeyer & Silva-Santos is a review article discussing Rho GTPase and their downstream signaling pathways in context of sepsis and septic shock, with a special focus on the potential targets for therapeutic interventions This is clearly a great choice for a review article, as the topic lies at the interface of conserved cellular mechanisms of Rho GTPase signaling and pathophysiological relevance in sepsis. I laud the author’s effort to identify not only this critical area of research, but more importantly highlight the major ideas in the field and the confusion and contradictions in the field literature, which have limited therapeutic advancement. To do so, authors have focussed on the studies conducted in experimental models and basic research and have presented a very thorough and well referenced overview of the field. I have a few concerns which I hope the authors will consider addressing.

Response to Reviewer #3 – General comments:  We appreciated your feedback and followed each of your comments, which helped us improve this review. As requested by the editor, all changes (except for adjustments in tables and figures) remain marked with the review tools from Microsoft Word. Thank you very much for your time and work on our submission.

Concerns (in order of appearance in text):

Comment #1:I like the framing of the title, but feel the message could be conveyed better if authors reframe the last portion into : ‘: What does basic research tell?’ or ‘:What have we learned from basic research?

Response to reviewer #3 – Comment #1: The last part of the title was reframed into  “what have we learned from basic research”. We agree with the reviewer that it sounds better than the original version.  

Comment #2: Please consider switching the order of paragraph 2 (Sepsis is defined…) and paragraph 1 (this review does…). Starting with paragraph 2 will give the article the big picture view of Sepsis as a critical life-threatening condition, followed by paragraph 1 which nicely defines the scope of the review.

 Response to reviewer #3 – Comment #2: In agreement with your suggestion, the second paragraph is now the first one, and the description of the aims were moved for the last part of the introduction. The following changes, performed in our submission, were done to address this comment: i) page 1, lines 33-40 – moved and reorganized on page 3, lines 110-115; ii) page 2, line 42, now the first sentence/paragraph.

Comment #3: The schematic in Figure 1 is really well illustrated but authors may want to add a little bit more explanation on the what the arrows define, and the boxes indicate. It’s could be a bit hard to follow the concept and message of the schematic. Please explain this figure in more details in the figure captions. 

Response to reviewer #3 – Comment #3: We thank you for this comment. The caption was carefully revised. Also, we did improve the figure to better fit with the aims of the figure (page 3).  

Comment #4: This manuscript does a great job in field literature survey and is a very well-referenced text. However, in the introduction, line 78 onwards, where authors list out the regulatory roles of Rho proteins in different processes ranging from adhesion, polarity, motility (lines 78-81), it will be good to add one more line saying something like this “ Recent studies have also now suggested Rho-GTPase signaling pathways crosstalk with each other and are influenced by cellular mechanics, leading to self-organisation of the several dynamic cellular processes. (refer to following reviews like PMID: 29632270, 31999511, 27533896).

Response to reviewer #3 – Comment #4: That was a significant suggestion and contribution. Following your tip, we did include the sentence with minor changes. Thanks a lot!

Comment #5: In line 170, authors discuss inconclusive findings from ROCK inhibition on barrier function. Are there studies which look at the dose dependence of ROCK inhibition on this process, is there a switch from improvement to worsening based on the dose regime or mode of administration.

Response to reviewer #3 – Comment #5:  That is a critical question, but the answer is no, at least to our knowledge. The lack of dose- or concentration-response experiments is a limitation found in the in vivo (as stated in our review) and in the in vitro studies found in this theme. Additionally, we would like to share with the reviewer our perception that, perhaps, the divergent findings may depend on different timepoints of evaluation, which is another significant gap in studies using ROCK inhibitors in sepsis-associated models.  

Comment #6: Line 180, please correct to Cdc42 from 43.

Response to reviewer #3 – Comment #6: That was corrected.

Comment #7: Figure 2, please increase the font size on the schematic to make the signaling pathway more readable.

Response to reviewer #3 – Comment #7: The central part of Figure 2 was revised to allow the correct visualization of the text, structures, and arrows in the final format of the Cells

Comment #8: In the last section, I wonder if it’s possible to highlight the concerns with the existing studies (lines 565-570) unknown questions in a bullet point fashion. It will allow readers to carry home the distilled message of the entire review.”

Response to reviewer #3 – Comment #8: We agree with the suggestion and listed the topics in sequence, although we are not sure whether this format will be maintained or revised in the final version.